# Variations in Genetic Diversity of Invasive Species *Lithobates catesbeianus* in China

**DOI:** 10.3390/ani14091287

**Published:** 2024-04-25

**Authors:** Jiaqi Zhang, Chunxia Xu, Supen Wang, Siqi Wang, Yiming Li

**Affiliations:** 1Key Laboratory of Animal Ecology and Conservation Biology, Institute of Zoology, Chinese Academy of Sciences, 1 Beichen West Road, Chaoyang, Beijing 100101, China; zhangjiaqi2025@163.com (J.Z.); xuchunxia0413@sina.com (C.X.); frog2019@ahnu.edu.cn (S.W.); wangsiqi17@mails.ucas.ac.cn (S.W.); 2University of Chinese Academy of Sciences, Beijing 100049, China; 3School of Life Sciences, Institute of Life Sciences and Green Development, Hebei University, Baoding 071002, China

**Keywords:** *Lithobates catesbeianus*, invasive species, rapid evolution, population genomics, molecular ecology

## Abstract

**Simple Summary:**

This study investigates the genetic structure and population dynamics of invasive bullfrogs in China employing cutting-edge genomic technologies. By analyzing microsatellite loci and single nucleotide polymorphisms (SNPs), the study identifies three distinct genetic subpopulations of bullfrogs and explores their origins and dispersal pathways. Findings reveal the varying degrees of gene flow between the subpopulations and a severe bottleneck effect followed by a rapid population expansion. Comparison with American bullfrog populations highlights the differences in genetic diversity and suggests potential multiple introductions.

**Abstract:**

The introduction and subsequent range expansion of the American bullfrog (*Lithobates catesbeianus*) is part of a rising trend of troublesome biological invasions happening in China. This detrimental amphibious invasive species has strong adaptability. After its introduction and spread, it established its own ecological niche in many provinces of China, and its range has continued to expand to more areas. Previous studies recorded the introduction time of bullfrogs and calculated the changes in their genetic diversity in China using mitochondria, but the specific introduction route in China is still unknown. Expanding upon previous research, we employed whole-genome scans (utilizing 2b-RAD genomic sequencing) to examine single nucleotide polymorphisms (SNPs) and microsatellites within *Lithobates catesbeianus* to screen the genomes of these invasive amphibian species from eight Chinese provinces and two U.S. states, including Kansas, where bullfrogs originate. A total of 1,336,475 single nucleotide polymorphic loci and 17 microsatellite loci were used to calculate the genetic diversity of bullfrogs and their migration pathways. Our results suggest that the population in Hunan was the first to be introduced and to spread, and there may have been multiple introductions of subpopulations. Additionally, the genetic diversity of both the SNP and microsatellite loci in the Chinese bullfrog population was lower than that of the US population due to bottleneck effects, but the bullfrogs can adapt and spread rapidly. This study will offer crucial insights for preventing and controlling future introductions into the natural habitats in China. Additionally, it will assist in devising more precise strategies to manage the existing populations and curtail their continued expansion, as well as aim to improve clarity and originality while mitigating plagiarism risk.

## 1. Introduction

The rate of invasive species is accelerating worldwide, resulting in increasing negative economic impacts [1,2,3]. Biological invasion can cause serious damage to the ecosystem in the invasion area through declines in biodiversity and by influencing the gene pool of the local species through hybridization or introgression [4]. The loss of biodiversity is mainly due to the following factors: preying on native species [5], competition for resources [6], reproductive disturbance [7], and disease or carrying infectious agents [8]. In their natural habitat, endemic amphibian, reptile, and mammal species have a harmonious balance of relationships, which affects the integrity and longevity of the ecosystem [9,10]. The introduction of invasive species can lead to the disruption of these relationships.

As a nation with a rapidly expanding trade industry, China has become home to more than 620 invasive alien species. Regrettably, this surge in global commerce has also resulted in the discovery of 51 out of the 100 most endangered exotic species worldwide within the territory of China [11]; they are causing significant economic damage, with numbers set to continue increasing in the future. By the end of 2019, China had documented a total of 515 species of extant amphibians [12]. Among them, the American bullfrog (*Lithobates catesbeianus*) stands out as a representative invasive amphibian species, which is one of the world’s worst invaders [13]. The invasion of bullfrogs has led to the decimation of some native amphibian populations due to their high reproductive ability, high densities, and predation on native amphibians [14]. The main harm of bullfrogs to native species comes from predation [14]. Due to its large size and wide diet, adult bullfrogs can prey on many smaller species [15]. Bullfrogs are also capable of spreading the chytrid fungus *Batrachochytrium dendrobatidis*, contributing to the rapid decline of amphibian populations in various environments [16]. In addition, bullfrogs can harm native species through competition for resources [6,17] and breeding disturbances [18].

Bullfrogs are intentionally introduced for aquaculture or as pets [19], and the wild populations of bullfrogs are established by farm escapes or artificial releases [20,21,22]. According to the literature, the bullfrog was introduced into China through Cuba and Japan in the 1950s and spread to all parts of the country for aquaculture operations [23]. There were three periods of the large-scale farming of bullfrogs in China [23]. The first was in the early 1960s, when bullfrogs were first introduced into China from abroad and bred. The breeding industry was suspended for twenty years for a variety of reasons. The second period was in the early 1980s; with the reform and opening, bullfrog farming also gradually resumed. At this time, bullfrogs were sourced from the wild populations of bullfrogs found in the Hanshou area of the Hunan Province, and this population was advocated to the whole country for breeding [23]. The third period coincided with the development of the farming industry; this is when the number of corporate-scale bullfrog farms gradually increased and they all implemented better methods to stop the escape behavior of bullfrogs [24]. However, the transmission route of the bullfrog in China is only rarely documented, and there are almost no complete molecular data to support it.

In our previous work, *Cytochrome b* genotype analysis showed that bullfrogs could successfully invade China and establish a wild population even with a lower genetic diversity than the US population [25]. The results of the study on the genetic structure of the population show that the two haplotypes in China are the same as the population branch in the western United States, indicating that the Chinese bullfrog may have originated from the western United States. Chinese invasive bullfrogs exhibit a 60% decrease in neutral genetic variations compared to their US native counterparts [26]. Despite experiencing lower genetic diversity at both *Cytb* and microsatellite loci compared to the US population, the Chinese bullfrog population exhibited rapid adaptation and expansion [26]. The transmission route of bullfrog after its introduction to China is still unknown, and using mitochondrial markers such as *Cytb* may have limitations. This is because *Cytb* is a single marker and typically reflects only the maternal lineage. However, population genomics can provide more power by using multiple markers, and it can offer a broader range of information beyond just the maternal lineage.

To study the genetic structure of an invasive species across different populations, it is typically necessary to gather samples from multiple populations and then analyze the genetic data among them using molecular markers. Genomic biosurveillance, comprising genomics and other state-of-the-art molecular technologies like whole-genome scans, has progressed significantly. It can swiftly provide management agencies and stakeholders with vital insights into invasive organisms in both their native and non-native habitats [27]. However, current research on the invasive species Chinese bullfrog in China has solely relied on the *cytb* molecular marker. The results regarding the origins, differentiation, and migration pathways of Chinese bullfrog populations in China are still insufficient. In our study, we will utilize microsatellite loci and single nucleotide polymorphisms (SNPs) to jointly investigate the population genetics of invasive bullfrogs in China. This approach will provide fine-scale, cost-efficient information for elucidating the invasion process [27,28].

Determining the origins of non-native species invasions aids in understanding the pathways and means of invasion, offering clearer directives to mitigate the threats posed by amphibious invasive species in the affected regions [28]. The main contribution of this study is the use of genome-wide and microsatellite loci to analyze the dispersal routes and population differentiation of the introduced alien bullfrogs into China, as well as the differences in genetic diversity between those bullfrogs originating from America. In this study, the bullfrog population was divided into three subpopulations, but there were varying degrees of gene flow between the different subpopulations. The use of genomic data to study bullfrog dispersal and population delimitation is more accurate than other studies in the literature. This study marks a significant methodological advancement in analyzing invasive species populations through cutting-edge genomic technology, offering a fresh array of data directly applicable to management stakeholders. The findings will notably enhance predictive capabilities regarding the potential expansion and the formulation of biological control strategies.

## 2. Materials and Methods

### 2.1. Ethics Statement

The methods were designed based on the Good Experimental Practices adopted by the Institute of Zoology, Chinese Academy of Sciences, China. All experimental procedures and animal collection were approved by the Animal Ethics Committee at the Institute of Zoology, Chinese Academy of Sciences, China (permit number: IOZ10013). All methods were carried out in accordance with the relevant guidelines and regulations.

### 2.2. Study Area and Sampling for Bullfrogs and DNA Extractions

In total, 10 bullfrog populations were sampled during the nonbreeding seasons (June to September) of 2010–2015, with 10 individuals collected from each population. Samples from two American populations in Kansas and California which were provided by the American Museum (Sternberg Museum and the University of California, Berkeley Museum of Vertebrate Zoology); eight Chinese populations were sampled in Zhejiang, Yunnan, Anhui, Tibet, Tianjin, Hunan, Shandong, and Sichuan (see Appendix A and Figure 1 for details). Frogs were systematically collected to cover the habitat diversity found at each site (e.g., streams, ponds, farmland, etc.). A total of 100 specimens were randomly selected from ten individuals per site for genotyping at the RAD and microsatellite loci. The captured frogs were released alive after clipping off the tip of the third toe of the right foot. The sampled toe tips were preserved in 95% ethanol and stored at −20 °C in the laboratory. Genomic DNA was extracted from the bullfrog tissue using an animal tissue extraction kit (TransGen MagicPure^®^ Genomic DNA Kit, Beijing, China).

### 2.3. Microsatellite Genotyping

The bullfrog populations were genotyped using 17 microsatellite loci (GenBank sequences to calculate genetic diversity and genetic structure [29]: AY323934, AY323931, AY323932, AY323930, AY323929, AY323928, HQ439092, HQ439093, HQ439094, HQ439096, HQ439097, AB911223, AB911228, AB911299, AB911231, AB911236, and AB911222. Primer sequences and the procedures used during DNA amplification were based on previously published data [30,31] (Appendix A). All primers were tagged with 5′-fluorescein bases (TAMRA, FAM, or HEX). The amplification conditions consisted of an initial denaturation at 94 °C for 3 min followed by 35 cycles of 10 s at 94 °C, 30 s at the annealing temperature (Appendix A), and 30 s at 72 °C and a final 10 min extension at 72 °C. Then, the PCR products were separated by 2% agarose gel electrophoresis. After amplification, the PCR products underwent resolution utilizing an ABI PRISM 377 DNA Sequencer (Applied Biosystems, California, CA, USA). Subsequently, the microsatellite fragments were scored using GENESCAN version 3.7 (Applied Biosystems) and GeneMarker version 1.71 (SofGenetics, Great Falls, VA, USA).

### 2.4. Genome Sequencing, Alignment, and SNP Calling

The DNA concentration and quality were assessed using a NanoDrop 2000 spectrophotometer (Thermo Fisher Scientific, Waltham, MA, USA). Subsequently, each DNA sample was adjusted to a concentration of 500 ng·μL^−1^ with a volume of 50 μL. For use in subsequent polymerase chain reactions (PCRs), the DNA was further diluted to a final concentration of 2.5 ng·μL^−1^. RAD-seq libraries were constructed following previously established protocols with minor adjustments [32]. Briefly, genomic DNA underwent a 5 min incubation at 37 °C with 20 U of Alf I restriction endonuclease (New England Biolabs, Ipswich, MA, USA) in a 50-μL reaction mixture. Subsequently, individually barcoded P1 adapters were ligated to the Alf I restriction site for each sample. Following this, samples were pooled in proportional amounts and sheared to an average size of 500 bp using a Bioruptor (Diagenode, Liège, Belgium). Sequencing libraries were constructed, accommodating a total of 24 samples per library. Size selection for fragments ranging from 450 to 550 bp was performed on a 2% agarose gel. The libraries underwent blunt end-repair, and a 3′-adenine overhang was added to each fragment. P2 ligation, incorporating unique Illumina barcodes (San Diego, CA, USA) to each library, was then carried out. PCR amplification of the libraries was conducted for 16 cycles (98 °C for 2 min; 16 cycles at 98 °C for 30 s, 60 °C for 30 s, and 72 °C for 15 s; and 72 °C for 5 min) using Phusion high-fidelity DNA polymerase (New England Biolabs, Ipswich, MA, USA), followed by column purification. Finally, sequencing was performed using a HiSeq 2500 system (Illumina) with 150 bp paired-end reads. We compared the sequenced genomic data with the bullfrog genome sequence RCv2.1 (BioProject PRJNA285814) in the NCBI database as the reference genome using samtools v1.11 software [33]. The alignments were filtered to remove low-quality mappings, PCR duplicates, and other artifacts. BamDeal v0.24 was used to calculate the sequencing depth and coverage of the samples. The HaplotypeCaller and GenotypeGVCFs modules of GATKv4.1.9.0 were used to detect SNPs among the samples [34]. To reduce false positive results in the subsequent analysis, the low quality alignments were removed (QUAL < 30.0, QD < 2.0, MQ < 40.0, FS > 60.0, SOR > 3.0, MQRankSum < −12.5, and ReadPosRankSum < −8.0) (https://gatk.broadinstitute.org/, accessed on 28 May 2022). Only the SNP polymorphism loci that passed the quality control conditions were retained for the subsequent analysis.

The SNP with minor allele frequency MAF > 0.05 were removed using vcftools V0.1.16 [35]. The filtered SNP data were used for the subsequent population structure analysis, including the phylogenetic tree and principal component analysis. We used RAD-seq to identify 275 genomic loci and candidate genes associated with the top 1% *Fst* values in the different subpopulations of the bullfrog.

### 2.5. Data Analyses

#### 2.5.1. Microsatellites

MICRO-CHECKER 2.2.3 was applied to quantify the scoring errors resulting from factors such as the large allele dropout [31], stuttering, or null alleles. GENEPOP version 4.0 was employed to test the linkage disequilibrium and Hardy–Weinberg equilibrium [36]. We applied Bonferroni corrections when performing multiple comparisons [37].

GenAlEx 6.5 was applied to quantify the expected heterozygosity (He), observed heterozygosity (Ho), mean number of alleles (Na), and the pairwise population differentiation coefficient (*Fst*) for each population [38]. Migration rates were estimated using the MIGRATE-N 3.2.7 program [39]. The approximate Bayesian computation (ABC) method was used to explore the hypothetical scenario of the bullfrog invasion of China, as implemented in DIYABC 1.0 software [40].

STRUCTURE was used to examine the genetic structure of the sampled bullfrogs [41]. The optimal number of clusters (the best K) was determined using the Evanno method [42] and was implemented in STRUCTURE HARVESTER [43]. STRUCTURE 2.3.4 was used to calculate the K value of the populations. We used the admixture model and correlated allele frequency model. STRUCTURE was run with 10 repetitions of 1,000,000 iterations of the MCMC simulation, following a burn-in of 200,000 iterations at K = 1–10.

#### 2.5.2. SNP Data

Principal component analysis (PCA) was performed on the bullfrog populations using GCTA V1.93.2 [44]. The first and second principal component results were plotted using R v4.1.0. Discriminant Analysis of Principal Components (DAPC) in the Adegenet v2.1.1 package [45] in R was used to analyze the SNP data to determine the population across each site. ADMIXTURE V1.3.0was used to estimate the ancestral composition of each individual by the cross-validation (CV) procedure [46]. To determine the optimal number of ancestral populations, the population structure K was assumed to be from 1 to 10. The visualization of these analyses was implemented in the Adegenet v2.1.1 package [45].

The DiveRsity v1.9.90 package in R was used to calculate the migration rates and the gene flow patterns among the invasive populations to determine how closely related they were when introduced [47]. We used DiveRsity to calculate the genetic diversity and differentiation statistics (Nm), as well as bootstrapped 95% confidence intervals for pairwise comparisons between the populations and the estimation of directional migration rates among the populations. We constructed a population phylogenetic tree for all samples by linking the homozygote of the SNP site and covered in all samples using FastTree 2.1.11 software using the neighbor-joining algorithm, with default values for the parameters [48].

The *Fst* can effectively express the degree of differentiation between the populations. We used vcftools V0.1.16 to calculate the pi values of the subpopulations using default parameters [35]. The interval with a higher differentiation between the populations was found by calculating the value between each population and combining the difference in genetic diversity to identify the variable loci.

## 3. Results

### 3.1. Microsatellite Genetic Structure in Chinese Bullfrog Populations

After using MICRO-CHECKER to analyze our data, we did not detect any null alleles or scoring errors. We then conducted a Bonferroni correction and found no evidence of linkage disequilibrium among the loci and populations with a *p*-value greater than 0.05. Furthermore, we observed no significant deviations from Hardy–Weinberg equilibrium in either the populations or the loci, even after applying Bonferroni corrections (*p* > 0.05). It is worth noting that all the loci were found to be polymorphic in every population analyzed. Based on the ΔK value, the most likely structure clustering was K = 3, and the probability of the clustering decreased when the K value increased further. Figure 2 shows the admixture frequency of the populations (also see Appendix A). The bullfrog population in China consisted mainly of three clusters. The first group included samples from two American sites. The second group included samples from Zhejiang, Yunnan, Shandong, and Sichuan Provinces. The third group of samples were from Anhui, Xizang, Hunan, and Tianjin Provinces.

The mean expected heterozygosity (He) and mean observed heterozygosity (Ho) of the microsatellites for American populations were 0.59 ± 0.05 and 0.62 ± 0.07, respectively, and those for Chinese populations were 0.57 ± 0.03 and 0.69 ± 0.04. The mean number of alleles (Na) was 6.56 ± 0.14 for the American populations and 4.79 ± 0.43 for the Chinese populations. Overall, the American populations had a higher microsatellite allelic richness (Na) than the Chinese populations (one-way ANOVA, F = 25.44, *p* < 0.05). There were no significant differences in observed heterozygosity (Ho) and expected heterozygosity (He) between the American and Chinese bullfrog populations (F = 0.1778, *p* = 0.685 and F = 4.37, *p* = 0.069) (Table 1). The average genetic differentiation coefficient (*Fst*) was 0.213 ± 0.08 for the microsatellites with 32/45 significant pairwise values in 10 populations (Appendix A).

### 3.2. Genome Sequencing and SNP Calling

A total of 508.97 GB of raw data were obtained in this project, with an average of 5.09 GB of data per sample, and 93.33% of the bases have achieved Q30 quality score (Appendix A). The average depth of the RAD in each sample sequencing is approximately 0.8×, and the coverage of the regions with a sequencing depth above 5× is approximately 5% (Appendix A).

We preliminarily obtained 6,8407,670 single-nucleotide polymorphic loci, which contained many regions with low depth or that had not been measured in many samples. We removed the SNP loci with low reliability, including the loci for which the occurrence rate was less than 75% in all samples and the MAF was less than 0.05 [49]. A total of 1,336,475 single nucleotide polymorphic loci were used for the subsequent analysis. The statistics summarizing the genetic diversity between the bullfrog populations revealed that each population contained private alleles (Table 1). The Kansas population contained the most private alleles and the Tianjin Province contained the least; the American population had more unique alleles than the Chinese population (Mann–Whitney U test: df = 9, Z = 2.089, *p* = 0.037).

The results of the principal component analysis (PCA) indicated genetic similarity between the Kansas and California populations, which we named subgroup 1. The samples from the Chinese invasion area were divided into two genetic subgroups (Figure 3). The samples from Anhui, Tianjin, and Xizang Provinces were grouped together, and referred to as subgroup 2. The samples from the other five sites, including Shandong, Sichuan, Hunan, Yunnan, and Zhejiang, belonged to the same genetic subgroup, referred to as subgroup 3. These results indicated that there was genetic dissimilarity between Chinese populations and American populations according to PC2 (Figure 3C). Additionally, the Chinese populations were divided into two subgroups according to PC1 (Figure 3B). However, one sample from Kansas was found within the Chinese subgroup. We calculated the population structure from K = 1 to K = 10 using the software STRUCTURE, and the results showed that when K = 3, the clustering results were consistent with the PCA results (Figure 4 and Appendix A).

The relative directional migration rates between the bullfrog populations are shown as a network (Figure 5). The results showed that the two American populations were grouped together, and the Chinese populations were classified into subgroups. The samples from Anhui, Tianjin, and Xizang Provinces were grouped together, and the samples from Shandong, Sichuan, Yunnan, and Zhejiang Provinces were grouped together. In addition, the bullfrog population in Hunan Province was between the two subgroups. The results of the migration rate were similar to those of the principal component analysis.

We used a total of 4247 homozygous SNPs, which have coverage in all locations and are homozygous in all samples to construct the phylogenetic tree (Figure 6). Using the phylogenetic tree, it was possible to divide all samples into five branches, with the American bullfrog populations represented by the KS and CA populations belonging to one branch; the AH, TJ, and XZ populations belonging to one branch; and the SD, ZJ, SC, YN, and HN populations belonging to one branch. except for the populations from Shandong and Anhui provinces, the populations from other regions were mixed with other populations. The results of the phylogenetic tree were consistent with the results of the relative directional migration rates between the bullfrog populations and principal component analysis.

The pairwise genetic distance (*Fst*) analysis revealed differences among the three subpopulations. In detail, there was a minimal difference between the American bullfrog subpopulation 1 and Chinese subpopulation 3 (*Fst* = 0.1159), a substantial difference between the American bullfrog subpopulation 1 and Chinese subpopulation 2 (*Fst* = 0.1448), and a relatively minor difference between the Chinese subpopulation 2 and subpopulation 3 (*Fst* = 0.1022). All *Fst* values were between 0.10 and 0.15, indicating a moderate level of genetic differentiation among the three subpopulations. In addition, we calculated Tajima’s D and pi values for each of the three subpopulations and calculated the difference in the SNP density between the two from the difference in pi values (Figure 7). The results indicated that the subpopulation 3 was exposed to a stronger positive selection, and both the subpopulation 2 and subpopulation 3 may have experienced a period of rapid population expansion following a bottleneck effect. Tajima’s D values for most windows in the subpopulation 1 were greater than 1, indicating that the bullfrogs were subject to a negative selection pressure from the environment; this is consistent with the subpopulation 1 being a natural population of the American bullfrogs.

All compared genetic loci included the region where the nad1 gene is located. The region where nad1 is located is above scaffold KX686108.1, and in this region, the subpopulation 1 exhibited a large separation from the subpopulation 3 with an *Fst* value of 0.53, the subpopulation 2 had an *Fst* value of 0.39, and the subpopulation 1 had an *Fst* value of 0.21 (Appendix A). 

## 4. Discussion

Identifying the sources and transmission pathways of invasive species is essential to minimize the damage caused by their spread and to protect native species [28,50]. For instance, in 2021, Resh’s study of the Northern snakehead in the United States using a genomic approach provided evidence that this harmful exotic species was likely to have come from the Yangtse River in China [28]. In another example, a study on the genetic diversity of the invasive species bullfrog in China showed that the genetic diversity in the Chinese bullfrog population undergoes adaptive selection to varying degrees of environmental pressures in different subgroups, which enables better adaptation to the new environment in the invaded area [29]. The research quantified the relative importance of propagule pressure and hunting pressures on the genetic variation of the bullfrog populations and found that insular populations have a greater genetic variation than their mainland counterparts.

The use of genomic data from the native range of invasive species for comparison with species in the invaded area facilitates ecological and evolutionary studies. Genomic data from the native sites of the invasive species are beneficial for ecological and evolutionary studies because they allow researchers to compare the responses of the invasive species to their native and introduced environments and to determine the success of the invasion [28,51,52]. For example, a study provided evidence using the genomic methods of genetic changes in two invasive goby species during the invasion period [53]. The study results showed that a higher proportion of functional loci experienced divergent selection for round gobies, suggesting that an increased evolutionary potential in invaded ranges may be associated with the greater invasion success of round gobies [53]. Future studies should also incorporate functional genetic markers to better assess the evolutionary potential for the improved conservation and management of species.

The Chinese bullfrog population is currently experiencing a bottleneck effect, and genetic diversity is being subjected to positive selection. Our results indicated that the genetic diversity of both microsatellites and SNPs was lower in the Chinese populations than in the American populations, providing evidence that bullfrogs with low genetic diversity can successfully invade. Based on the research conducted by Tsutsui et al., it was revealed that the invasive Argentine ant (*Linepithema humile*) experienced a decline in genetic diversity due to a bottleneck effect during the invasion process. Despite this reduction, the ant species was still able to successfully establish colonies and spread within the invaded area, demonstrating its adaptability and resilience [54]. Three factors may have largely contributed to the extremely low genetic diversity of the bullfrogs successfully established in China. First, the high fertility of the bullfrogs may allow them to survive demographic bottlenecks, as evidenced by the invasive bullfrogs in Europe, where the most invasive populations originated from fewer than six females [55]. Second, when invasive species are introduced with a small population size, they are subject to genetic drift and increased inbreeding, which can lead to an accumulation of deleterious mutations and reduced fitness. However, the increased frequency of recessive homozygotes in such small populations can enhance the effectiveness of purging against deleterious mutations. Consequently, the potential for purging inbreeding depression is heightened, thereby enabling the invasive species to better adapt to new environments [56]. Finally, there is a high reproductive pressure due to repeated escape from commercial populations [24].

Through selection elimination analysis, we discovered that the Nad1 (Nicotinamide Adenine Dinucleotide Dehydrogenase Subunit 1) gene, which has important functions in the tricarboxylic acid cycle and respiratory chain [57], exhibited high variability among different bullfrog subpopulations. The sequence information of Nad1 intron 2 is informative, making it suitable for conducting phylogenetic studies at higher taxonomic levels [58]. A study on frogs (Neobatrachia) revealed variations in the evolutionary rates of mitochondrial genomes among different lineages of frogs. Compared to their non-Neobatrachian relatives, Neobatrachians exhibit accelerated evolutionary rates, a phenomenon that originated with the emergence of Neobatrachia [59]. The variation in Nad1 in the three subpopulations coincided with the results of our population clustering. The genetic variation in the Nad1 gene may have some effect on the successful invasion and spread of the bullfrog populations, and the Nad1 gene is associated with respiration and metabolism, suggesting that Chinese bullfrog populations may have undergone rapid genetic evolution to adapt to different habitats.

The bullfrog has invaded China and established wild populations for over half a century [60]. However, it is difficult to accurately reconstruct the migration of bullfrog populations based solely on the literature and records. According to our results, the bullfrog population clustered into three subgroups. However, in the SNP analysis, one sample of the Kansas population was in the Chinese subgroup in PCA, while in the structure analysis, samples were different from Chinese populations. Possible reasons for this inconsistency include the following: The PCA analysis might capture subtle genetic variations that are not adequately represented in the population structure analysis [61]. Additionally, the PCA analysis might focus on specific genetic markers or regions that exhibit the different patterns of variation compared to the broader genomic background represented in the structure analysis [62]. Our results demonstrate geographic isolation exists within both bullfrog subgroups in China. For instance, notable geographical barriers between the Tibetan and Tianjin populations in the subgroup 2 include the Qinghai–Tibet Plateau, with an average elevation exceeding 4000 m; the Qinling Mountains, extending eastward from Tibet to north of Tianjin; and the Yellow River and Yangtze River, where water flow velocity and channel width may impede amphibian movement. Therefore, it is unlikely for frogs from different subgroups to disperse and expand their populations solely through their own migration efforts. Based on the analysis of the migration rate and direction, we speculate that the wild bullfrog population in Hunan was dispersed to different parts of China and formed different subpopulations during the second extensive breeding period. We can infer that the bullfrogs in Hunan represent the most ancient genetic population and their spread did not follow the expected geographic pattern. This is because their distribution was largely influenced by human activities such as artificial breeding and transportation, which led to the clustering of the populations without a significant correlation with distance or geographical location. The genetic subgroup 3 of Chinese bullfrogs showed a close genetic relationship with the American population, suggesting a possible introduction to China prior to significant genetic differentiation. In contrast, the greater genetic differences observed between the bullfrog genetic subgroup 2 and the American populations may have been due to a second introduction or a genetic bottleneck that facilitated better adaptation to the new environment. Our findings also revealed that the Chinese bullfrog subpopulations, particularly the subpopulation 3, underwent a severe bottleneck effect, followed by a rapid expansion in the population size.

The results were consistent with those of the previous studies we conducted in which we sequenced the mitochondrial *Cytb* gene region in 510 samples from wild and farm individuals across China and compared them to the populations within their local range, showing that the number of haplotypes in the Chinese population (N = 2) was much smaller than the number of haplotypes in their native range (N = 41), as well as the number of haplotypes in Europe (N = 5) [25]. The path of the introduction of the European bullfrogs was recorded at least 25 times from the 1930s to the 1990s [20], while the introduction of the Chinese bullfrogs was recorded only once [24]. The differences in the genetic diversity of the invasive bullfrogs between Europe and China were consistent with the hypothesis that multiple introductions can mitigate the loss of genetic diversity in invaders [63].

Based on the findings of the study, there are several potential strategies for controlling the alien species bullfrog. Targeted management and control efforts: focusing control measures on areas with high genetic diversity or where significant gene flow occurs can potentially be more effective in limiting the spread of the species. Genetic monitoring and surveillance: Continued genetic monitoring and surveillance of bullfrog populations can provide valuable information on population dynamics, including changes in genetic diversity and the emergence of new subpopulations. This information can guide adaptive management strategies and early intervention measures to prevent further spread. Public awareness and education: Increasing public awareness about the ecological and economic impacts of invasive species like the bullfrog can foster support for control efforts. Education programs can also help prevent unintentional introductions and encourage responsible pet ownership practices to reduce the risk of further spread.

## 5. Conclusions

The study presented here underscores the importance of utilizing advanced genomic techniques to unravel the intricate dynamics of invasive species using the Chinese bullfrog as a case study. Through the integration of microsatellite loci and SNP analysis, we gained valuable insights into the population genetics and invasion pathways of the bullfrogs in China. Firstly, our findings elucidate the genetic structure of the bullfrog populations in both their native American range and their introduced Chinese habitats. We identified three distinct genetic subgroups within the Chinese bullfrog population, with varying degrees of genetic differentiation from their American counterparts. This suggests complex invasion dynamics, possibly involving multiple introduction events and the subsequent adaptation to the new environment. Secondly, our study highlights the role of human activities, such as commercial breeding and transportation, in shaping the distribution and genetic composition of the invasive bullfrog populations. The clustering of the populations without strict geographic correlation underscores the influence of anthropogenic factors in facilitating the dispersal and establishment of the invasive species. Thirdly, the observed bottleneck effect followed by a rapid population expansion in the Chinese bullfrog subpopulations emphasizes the resilience and adaptability of the invasive species to new environments. Despite experiencing a reduction in genetic diversity, the bullfrogs with low genetic variation were still able to successfully establish and proliferate, highlighting the importance of considering the demographic and genetic factors in invasive species management strategies. Lastly, our study contributes to the broader understanding of invasion biology by showcasing the power of genomic tools in unraveling the origins, transmission pathways, and adaptive mechanisms of invasive species. By providing fine-scale genetic data, our research offers valuable insights for the effective management and mitigation of the ecological and economic impacts associated with invasive species invasions.

In conclusion, the integration of genomic approaches offers a robust framework for studying invasive species dynamics, with implications for both scientific research and practical management strategies aimed at preserving the native biodiversity and ecosystem integrity.

## Figures and Tables

**Figure 1 animals-14-01287-f001:**
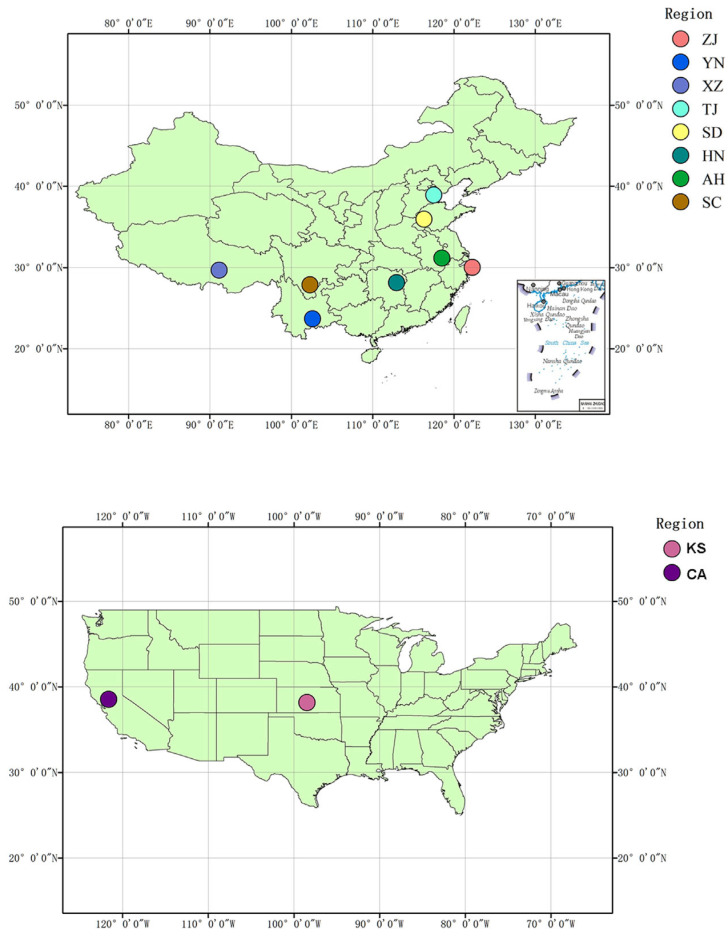
Collection locations of the *Lithobates catesbeianus* individuals in China and the United States. See Appendix A for details. Population abbreviations are as follows: KS, Kansas; CA, California; ZJ, Zhejiang; YN, Yunnan; AH, Anhui; XZ, Tibet; TJ, Tianjin; HN, Hunan; SD, Shandong; SC, Sichuan.

**Figure 2 animals-14-01287-f002:**
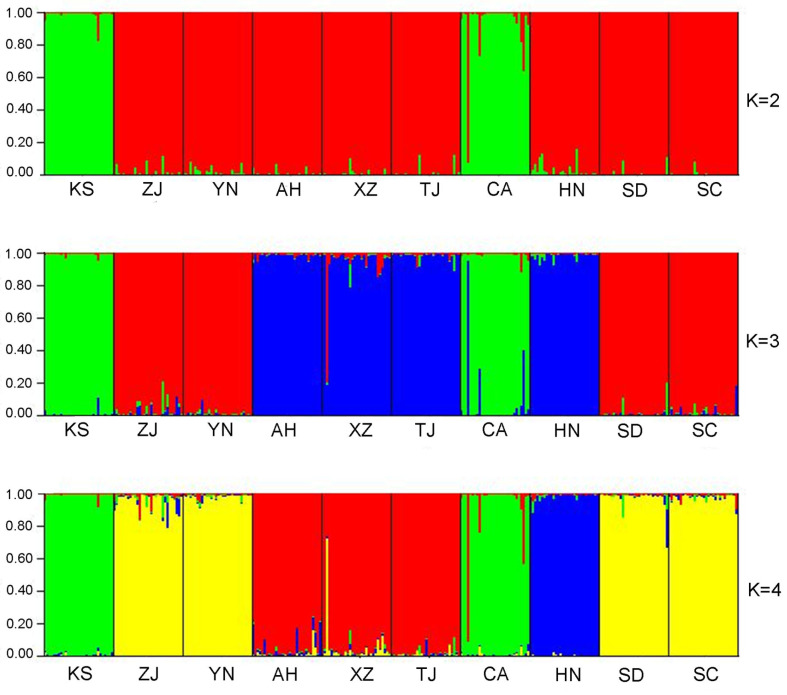
Clustering results in different locations from STRUCTURE using microsatellite data from *Lithobates catesbeianus* populations. Population abbreviations are as follows: KS, Kansas; CA, California; ZJ, Zhejiang; YN, Yunnan; AH, Anhui; XZ, Tibet; TJ, Tianjin; HN, Hunan; SD, Shandong; SC, Sichuan.

**Figure 3 animals-14-01287-f003:**
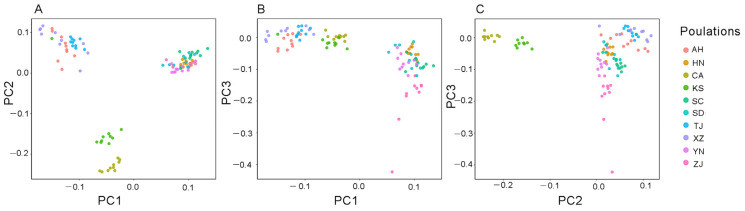
(**A**) Components 1 and 2, (**B**) components 1 and 3, and (**C**) components 2 and 3. Principal component analyses using the SNP data from the *Lithobates catesbeianus* populations. Population abbreviations are as follows: KS, Kansas; CA, California; ZJ, Zhejiang; YN, Yunnan; AH, Anhui; XZ, Tibet; TJ, Tianjin; HN, Hunan; SD, Shandong; SC, Sichuan. The three principal components explain 53.25% of the variation in the datasets.

**Figure 4 animals-14-01287-f004:**

Clustering results in different locations from STRUCTURE using SNP data from *Lithobates catesbeianus* populations. Population abbreviations are as follows: KS, Kansas; CA, California; ZJ, Zhejiang; YN, Yunnan; AH, Anhui; XZ, Tibet; TJ, Tianjin; HN, Hunan; SD, Shandong; SC, Sichuan.

**Figure 5 animals-14-01287-f005:**
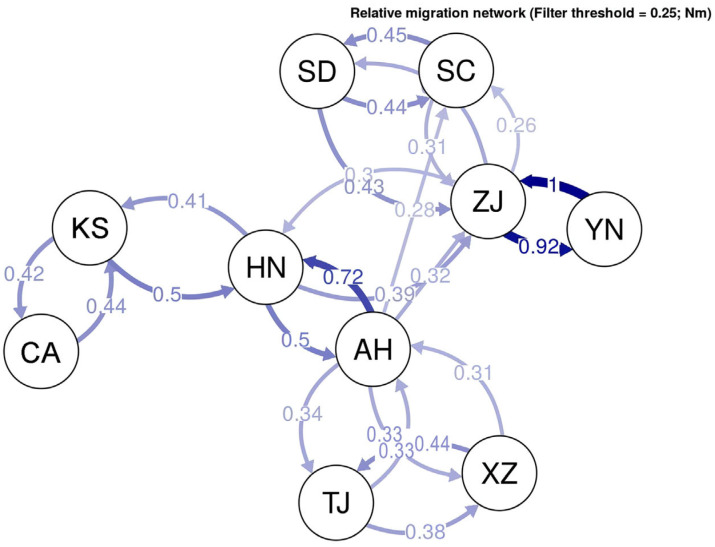
Relative migration network of the *Lithobates catesbeianus* populations. Each node represents a locality, and the arrows indicate the direction of gene flow, with the relative strength of the flow indicated by the bootstrap support value, as well as the shading and thickness of each connecting line. Population abbreviations are as follows: KS, Kansas; CA, California; ZJ, Zhejiang; YN, Yunnan; AH, Anhui; XZ, Tibet; TJ, Tianjin; HN, Hunan; SD, Shandong; SC, Sichuan.

**Figure 6 animals-14-01287-f006:**
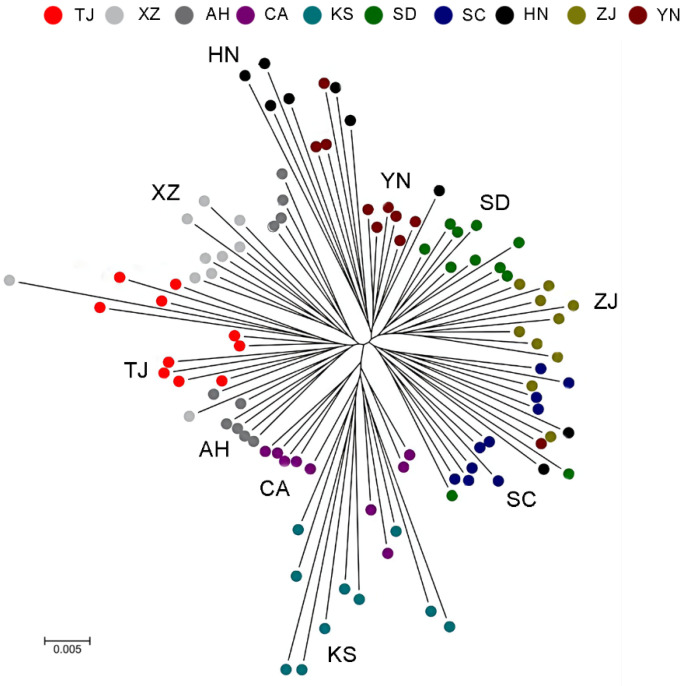
Phylogenetic tree of *Lithobates catesbeianus* in different populations. Nodes represent common ancestors; branches represent evolutionary relationships; branch lengths represent the magnitude of differences during the evolutionary process. Population abbreviations are as follows: KS, Kansas; CA, California; ZJ, Zhejiang; YN, Yunnan; AH, Anhui; XZ, Tibet; TJ, Tianjin; HN, Hunan; SD, Shandong; SC, Sichuan.

**Figure 7 animals-14-01287-f007:**
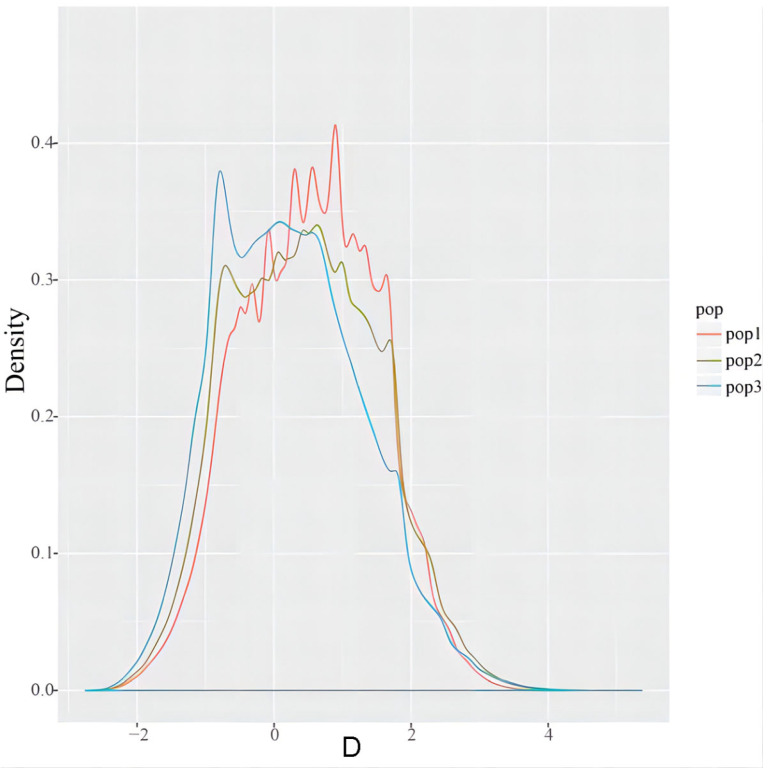
Tajima’s D statistical analysis of *Lithobates catesbeianus* in different subpopulations. The red line (pop1) in the figure represents the Tajima’s D value of group 1, the green line (pop2) represents the Tajima’s D value of group 2, and the blue line (pop3) represents the Tajima’s D value of group 3. The value of most windows in group 1 is greater than 1; group 2 and group 3 have a peak value near D = −1.

**Table 1 animals-14-01287-t001:** Genetic diversity of *Lithobates catesbeianus* in different populations. He = expected heterozygosity. Ho = observed heterozygosity. Na = mean number of alleles.

	SNP Data			Microsatellite Data		
Location	Private Alleles	Ho	He	Na	Ho	He
Kansas	17,304	0.13	0.12	6.7 ± 0.87	0.62 ± 0.07	0.6 ± 0.04
California	15,581	0.12	0.14	6.411 ± 1	0.62 ± 0.07	0.57 ± 0.06
Hunan	9207	0.10	0.12	5.47 ± 0.63	0.68 ± 0.07	0.57 ± 0.06
Yunnan	5239	0.06	0.11	5.23 ± 0.55	0.7 ± 0.05	0.61 ± 0.05
Anhui	6767	0.07	0.10	5.05 ± 0.67	0.79 ± 0.06	0.6 ± 0.04
Tibet	5117	0.09	0.09	4.88 ± 0.54	0.66 ± 0.08	0.53 ± 0.06
Tianjin	4330	0.08	0.08	4.76 ± 0.48	0.66 ± 0.06	0.58 ± 0.04
Zhejiang	5382	0.10	0.11	4.47 ± 0.35	0.68 ± 0.06	0.6 ± 0.04
Shandong	7408	0.10	0.09	4.35 ± 0.49	0.65 ± 0.09	0.52 ± 0.06
Sichuan	9809	0.08	0.10	4.12 ± 0.48	0.73 ± 0.06	0.58 ± 0.04

## Data Availability

The 2b-RAD sequencing data of *Lithobates catesbeianus* presented in this study are deposited in the NCBI BioSample Database under accession numbers SAMN31278454 through SAMN31278553 in BioProject PRJNA890400. The microsatellite data of *Lithobates catesbei* are provided in the Appendix A.

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
