# Peer review of "Variations in Genetic Diversity of Invasive Species Lithobates catesbeianus in China"

_animals, 2024, doi:10.3390/ani14091287_

Round 1
Reviewer 1 Report
Comments and Suggestions for Authors This study analyzes the non-indigenous populations of bullfrog collected from a wide area in China by microsatellite and RAD-seq analyzes. The results indicated the genetic population structure more cleary than previous studies. In particular, it is a new finding that the Chinese populations were differentiated from the North American populations, and were further divided into two groups within China. It is important because the finding sheds light on the distribution expansion process of alien populations. Although the methods and literature citations are appropriate, revisions are necessary. 1. Although the authors described K=3 is the best in MS and SNP analyses, it is necessary to describe the basis that K=3 is the best. (For example, delta K among number of clustering) 2. In SNP analyses, one of the KS samples was in China subgroup in PCA. On the other hand, all of the KS samples were different from Chinese populations in Fig 4. Why? Differences in results must be described. 3. In L314-318, abbreviations of localities are different from Fig 1-5 and other part of maintext. Moreover, the description of the results is different from the Fig 6. These careless descriptions raise doubts about the accuracy of this study. Please correct it. And please use the same color for each point as in the other figures.4. In discussion, the authors may need to explain the geographic distribution of the two subgroups and provide information that supports the hypothesis that the subgroups correspond to the first and second extensive breeding periods.
Author Response
Point-by-Point Responses to Reviewer Comments
Reviewer #1: This study analyzes the non-indigenous populations of bullfrog collected from a wide area in China by microsatellite and RAD-seq analyzes. The results indicated the genetic population structure more cleary than previous studies. In particular, it is a new finding that the Chinese populations were differentiated from the North American populations, and were further divided into two groups within China. It is important because the finding sheds light on the distribution expansion process of alien populations. Although the methods and literature citations are appropriate, revisions are necessary.
- Although the authors described K=3 is the best in MS and SNP analyses, it is necessary to describe the basis that K=3 is the best. (For example, delta K among number of clustering)
- In SNP analyses, one of the KS samples was in China subgroup in PCA. On the other hand, all of the KS samples were different from Chinese populations in Fig 4. Why? Differences in results must be described.
- In L314-318, abbreviations of localities are different from Fig 1-5 and other part of maintext. Moreover, the description of the results is different from the Fig 6. These careless descriptions raise doubts about the accuracy of this study. Please correct it. And please use the same color for each point as in the other figures.
- In discussion, the authors may need to explain the geographic distribution of the two subgroups and provide information that supports the hypothesis that the subgroups correspond to the first and second extensive breeding periods.
Our reply: Thank you for your overall positive comments on our findings. We have completely agreed with and taken these comments carefully. We have made changes as following:
- Now we validated that K=3 was optimal by calculating delta K using STRUCTURE HARVESTER. The results are presented in Supplementary Figure S1.
- We explained the differences in the SNP analysis results in the discussion (Lines 489-496, New line number). According to our results, the bullfrog population clustered into three subgroups. However, in the SNP analysis, one sample of the Kansas population was in the Chinese subpopulation in PCA, while in the structure analysis, samples were different from Chinese populations. Possible reasons for this inconsistency include: The PCA analysis might capture subtle genetic variations that are not adequately represented in the population structure analysis. Additionally, the PCA analysis might focus on specific genetic markers or regions that exhibit different patterns of variation compared to the broader genomic background represented in the structure analysis.
- We have re-edited this.
- In our discussion, we provided an explanation of how geographical factors within subgroups hindered the bullfrog population from freely dispersing and forming groups (Lines 504-511, New line number).
Our results demonstrate geographic isolation exists within both bullfrog subgroups in China. For instance, notable geographical barriers between the Tibetan and Tianjin populations in subgroup 2 include the Qinghai-Tibet Plateau, with an average elevation exceeding 4000 meters; the Qinling Mountains, extending eastward from Tibet to north of Tianjin; and the Yellow River and Yangtze River, where water flow velocity and channel width may impede amphibian movement. Therefore, it is unlikely for frogs from different subgroups to disperse and expand their populations solely through their own migration efforts.
Reviewer 2 Report
Comments and Suggestions for Authors
Dear authors,
The novelty of the content of the manuscript and its practical significance is undeniable. I have no objections to my own results and statistical methods. However, the title of the manuscript, chapter introduction and discussion raise many comments. I propose to clearly write the aim and formulate a hypothesis. Supplement Materials and Methods based on research samples. The Discussion will focus on comparing genetic analysis data for other species of invasive amphibians. After major revision, the manuscript may be reviewed again.

Author Response
Reviewer #2: The novelty of the content of the manuscript and its practical significance is undeniable. I have no objections to my own results and statistical methods. However, the title of the manuscript, chapter introduction and discussion raise many comments. I propose to clearly write the aim and formulate a hypothesis. Supplement Materials and Methods based on research samples. The Discussion will focus on comparing genetic analysis data for other species of invasive amphibians. After major revision, the manuscript may be reviewed again.
Our reply: Thank you for that you like our manuscript. We have completely agreed with and taken these comments carefully. We have made changes as following:
Line1-2 (New line number, same as above):Title of the manuscript is too detailed and does not accurately convey the essence of the work. The manuscript used frog specimens from the USA as well. And potential sources a priori follow from genetic analysis. Therefore, I propose the following option: "Variations in genetic diversity invasive species Lithobates catesbeianus in China"
Our reply: We accept this suggestion and will change the title to “Variations in genetic diversity invasive species Lithobates catesbeianus in China”
Line36:Delete. It's not needed here.
Our reply: We have already deleted this sentence.
Line 56-59: In their natural habitat, endemic amphibian, reptile and mammal species have a harmonious balance of relationships, which affects the integrity and longevity of the ecosystem [Thompson et al., 2016; Stepanova et al., 2021]. The introduction of invasive species can lead to disruption of these relationships.
Our reply: We have added the citation.
Line60-65: First, you need to add information on the number of amphibian species in China. Separately indicate the known number of invasive species. List in parentheses several species of the most common invasive amphibians in China. And then move on to the amphibian species under study. It will be logical and correct.
Our reply: We have added the current numbers of amphibian species and the number of alien species in China as background information in the introduction.
Line106-159: These paragraphs contain many methodological elements and should be moved to the Materials and Methods chapter. You can't leave it like this. However, the aim and objectives of the study have not been formulated. No hypothesis.
Our reply: We have re-edited this as suggested.
Line160:Transfer this figure to Materials and Methods.
Our reply: We have already moved the figure to the Materials and Methods.
Line169:Specify months
Our reply: We have added specific months.
Line170-171:It is not clear how many frogs there were from each point? The details cannot be understood from Figure and Table 1. It is necessary to provide a separate table with the characteristics of these individuals (gender, age, etc.).
Our reply: We have added the sampling number at each sampling site.
Line176:Which? Specify
Our reply: We have added the habitat types of the sampling sites.
Line403:This is the methodological part.
Our reply: We have already moved this part to the methodology section.
Line411-419: This part is more suitable for Introduction. These proposals can be applied by justifying the relevance and novelty of the research.
Our reply: We have already moved this part to the introduction section.
Line424-425: Delete. This is not relevant to the discussion of your results.
Our reply: We have already deleted this sentence.
Line426: No need to specify completely. You already have a link at the end of the sentence.
Our reply: We have removed the specific names of the authors.
Line432-433: This again makes more sense for the conclusion. But not for every paragraph.
Our reply: We have moved this sentence to the conclusion.
Line439: similarly.
Our reply: We have removed the specific names of the authors.
Line465-467: It’s also better to move it to the final part.
Our reply: We have moved this sentence to the conclusion.
Line473-480: You bring into the discussion a lot of comparisons with animals from different systematic groups. The focus should be on different species of invasive amphibians. I believe that enough publications have been devoted to the genetic issues of invasive frogs. Do you agree?
Our reply: We agree with your point. We have already replaced the specie in the discussion with more representative amphibians.
Line485-488: This is a repeat sentence. You already wrote about this in the Introduction (line 52).
Our reply: We have re-edited this.
Line497-504: You already mentioned this in the results. Why do you need a repeat?
Our reply: We have deleted the redundant description.
Line549-598: This is not a conclusion based on your research. You can't write like that.
Our reply: We have already rewritten the conclusion.
Line549-598: This paragraph should be rephrased to reflect comments made in the Discussion. Add original conclusions based on your research. In this form, it looks abstract. Especially on possible pathways of bullfrogs.
Our reply: We have already rewritten the conclusion.

Reviewer 3 Report
Comments and Suggestions for Authors
This study utilizes genome-wide and microsatellite loci to analyze the dispersal routes and population differentiation of introduced alien bullfrogs into China, as well as the differences in genetic diversity between those bullfrogs originating from America. The researchers identified that the population in Hunan was the first to be introduced and to spread, suggesting the possibility of multiple introductions of subpopulations. Furthermore, the genetic diversity in the Chinese bullfrog population was found to be lower than that of the US population, attributed to bottleneck effects. Overall, this study is of significant importance and merits publication.
I have a few general comments to enhance the manuscript:
1. While I appreciate the focus of this study on animal genetics and genomics, considering the invasive nature of the bullfrog species under investigation, it would be beneficial for the authors to elaborate on potential strategies for controlling the alien species based on the study findings. This discussion would not only add value to the manuscript but also engage invasive ecologists, thereby contributing to invasive species management efforts.
2. The manuscript highlights the identification of the spread route of the invasive bullfrog species in China. To enrich the ecological perspective of the study, I recommend the authors to explain the spread route discussion from an ecological standpoint. Moreover, providing insights into distinguishing between natural spread routes and those introduced by human activities would signify a significant advancement in this field.
Author Response
Reviewer #3: This study utilizes genome-wide and microsatellite loci to analyze the dispersal routes and population differentiation of introduced alien bullfrogs into China, as well as the differences in genetic diversity between those bullfrogs originating from America. The researchers identified that the population in Hunan was the first to be introduced and to spread, suggesting the possibility of multiple introductions of subpopulations. Furthermore, the genetic diversity in the Chinese bullfrog population was found to be lower than that of the US population, attributed to bottleneck effects. Overall, this study is of significant importance and merits publication.
I have a few general comments to enhance the manuscript:
- While I appreciate the focus of this study on animal genetics and genomics, considering the invasive nature of the bullfrog species under investigation, it would be beneficial for the authors to elaborate on potential strategies for controlling the alien species based on the study findings. This discussion would not only add value to the manuscript but also engage invasive ecologists, thereby contributing to invasive species management efforts.
- The manuscript highlights the identification of the spread route of the invasive bullfrog species in China. To enrich the ecological perspective of the study, I recommend the authors to explain the spread route discussion from an ecological standpoint. Moreover, providing insights into distinguishing between natural spread routes and those introduced by human activities would signify a significant advancement in this field.
Our reply: Good points, which improved the quality of the manuscript. We have taken the comments seriously, and make revisions as following:
- Thank you for your suggestion. Based on the research results, we have elaborated on potential strategies for controlling invasive species in detail at the end of the discussion (Line536-547, new line number).
Based on the findings of the study, there are several potential strategies for control-ling the alien species bullfrog. Targeted Management and Control Efforts: Focusing control measures on areas with high genetic diversity or where significant gene flow occurs can potentially be more effective in limiting the spread of the species. Genetic Monitoring and Surveillance: Continued genetic monitoring and surveillance of bullfrog populations can provide valuable information on population dynamics, including changes in genetic diversity and the emergence of new subpopulations. This information can guide adaptive management strategies and early intervention measures to prevent further spread. Public Awareness and Education: Increasing public awareness about the ecological and economic impacts of invasive species like the bullfrog can foster support for control efforts. Education programs can also help prevent unintentional introductions and encourage responsible pet ownership practices to reduce the risk of further spread.
- Thank you for your suggestion. We have incorporated discussions on the pathways of spread from an ecological perspective into the manuscript. (Line504-511, new line number).
Our results demonstrate geographic isolation exists within both bullfrog subgroups in China. For instance, notable geographical barriers between the Tibetan and Tianjin populations in subgroup 2 include the Qinghai-Tibet Plateau, with an average elevation ex-ceeding 4000 meters; the Qinling Mountains, extending eastward from Tibet to north of Tianjin; and the Yellow River and Yangtze River, where water flow velocity and channel width may impede amphibian movement. Therefore, it is unlikely for frogs from different subgroups to disperse and expand their populations solely through their own migration efforts.
Round 2
Reviewer 2 Report
Comments and Suggestions for Authors
Dear Authors,
I am satisfied with the correction of the manuscript. You have supplemented the manuscript with data and made corrections. The research material used was selected appropriately, as well as the statistical methods used to analyze it. Own results and their discussion in the Discussion are properly described and were compared with previous studies by other authors. I recommend it for the journal Animals.